# A time-varying geospatial model of habitat suitability for Japanese encephalitis virus vectors and vertebrate hosts in Australia

David H. Duncan[1,2*], Lucinda E. Harrison[3¤a], Abbey Potter[4], Craig Brockway[4], Kimberly L. Miller[4], Stephen L. Doggett[5], Rebecca Feldman[6], Peter J. Neville[6], Andrew F. van den Hurk[7], Cassie C. Jansen[8], Michaela Hobby[9¤b], Vicki Burns[9], Andrew Vickers[9], Nina Kurucz[10], Nick Golding[1,2,11�], Freya M. Shearer[2�]

1 Infectious Disease Ecology and Modelling, The Kids Research Institute Australia, Nedlands, Western Australia, Australia, 2 Melbourne School of Population and Global Health, University of Melbourne, Carlton, Victoria, Australia, 3 School of Mathematics and Statistics, University of Melbourne, Parkville, Victoria, Australia, 4 Environmental Health Directorate, Western Australia Department of Health, East Perth, Western Australia, Australia, 5 Department of Medical Entomology, NSW Health Pathology, Westmead Hospital, Westmead, New South Wales, Australia, 6 Community and Public Health Division, Department of Health, Melbourne, Victoria, 7 Public Health Virology, Public and Environmental Health Reference Laboratories, Queensland Health, Coopers Plains, Queensland, Australia, 8 Health Protection and Regulation Branch, Queensland Health, Brisbane, Queensland, Australia, 9 Health Protection and Regulation, Department for Health and Wellbeing, SA Health, Adelaide, South Australia, Australia, 10 Medical Entomology - Centre for Disease Control, Public Health Division, NT Health, Tiwi, Northern Territory, Australia, 11 Physics, Maths and Computing, University of Western Australia, Crawley, Western Australia, Australia

☉ These authors contributed equally to leading this work.
¤a Current Address: Infectious Diseases Data Observatory, University of Oxford, Oxford, UK
¤b Current Address: Department of Agriculture, Fisheries and Forestry, Australian Government, Export Park, Australia
* david.duncan@unimelb.edu.au

## Abstract

In the austral summer of 2021–2022, Australia experienced an unprecedented Japanese encephalitis virus (JEV) outbreak, with detections over 3000 km south of previous occurrences. Given the limited knowledge of JEV transmission ecology in Australia, we developed geospatial models of transmission risk to support the public health response. We created time-varying habitat suitability models for suspected mosquito vectors and ardeid hosts using month-scaled occurrence and covariate data from 2000–2023. Ardeid host presence-absence data were obtained from eBird and BirdLife Australia, with habitat suitability estimated using gradient-boosted regression tree models. A national dataset of *Culex* occurrences was compiled from mosquito surveillance records, literature, and biodiversity databases. Penalised logistic regression was used to model mosquito vector habitat suitability. Vector and host habitat predictions for the outbreak peak in February 2022 were rescaled using JEV infection locations in the public domain to create a combined habitat suitability surface. Our models aligned with detected JEV infections at the continental scale, highlighting transmission suitability across tropical northern Australia and major inland drainage basins in the East. Unlike existing models, we predicted lower

**Data availability statement:** The structured occurrence dataset of ardeid species comprised confirmed complete records obtained from eBird and Birdlife Australia as cited in text. The records can be downloaded for free. In the case of eBird, a user must establish an account and provide a project description available at https://support.ebird.org/en/support/solutions/articles/48000838205-download-ebird-data. Our eBird download referenced eBird Basic Dataset Version EBD_relAug2023. Equivalent data can be requested for free for research purposes from Birdlife Australia by filling in a request form available from https://birdata.birdlife.org.au/request-data-2025. The cross-validation dataset of Large Wading Birds from the Eastern Australia Waterbird Survey of UNSW can also be downloaded free of charge by signed in users at this page https://aws.ecosystem.unsw.edu.au/ or from links provided in text. The Culex vector records we obtained from biodiversity atlases can be recovered using the respective DOI from Atlas of Living Australia (https://doi.ala.org.au/doi/e9a7538c-ffa6-4a62-a269-c6e2858aae7d) and GBIF (Culex: https://doi.org/10.15468/dl.vdwtvb, Animalia target-group background: https://doi.org/10.15468/dl.a6vucc) or via the DOI link in their respective citations in the text. The mosquito occurrences used in the analysis included surveillance records obtained from third-party state and territory government institutions. We were unable to obtain permission from the institutions to publicly archive those data due to concerns about the sensitivity of trap locations on commercial and private property, however, an analysis ready version of each component of the final dataset has been lodged with each of the respective institutions. The component datasets can be requested from each institution detailed below via a data sharing agreement: New South Wales: Surveillance and Risk Unit, Environmental Health Branch, Health Protection, St Leonards, NSW. HSSG-EHBSurveillance@health.nsw.gov.au Northern Territory: Centre for Disease Control, Public Health Division, NT Health, Darwin. mailto:MedicalEntomologyRDH.THS@nt.gov.au Queensland: Health Protection and Regulation Branch, Queensland Health mailto:medicalentomology@health.qld.gov.au South Australia: Health Protection Programs, Public Health

suitability along the eastern seaboard, suggesting a delimiting effect of the Great Dividing Range. Our approach provides the most comprehensive and temporally dynamic models for JEV hosts and vectors in Australia, with a significantly larger vector dataset than previous studies. The novel method of rescaling host and vector outputs into a combined surface offers new insights into JEV transmission risk. Favourable conditions were repeated in 2023 with few detected infections, emphasising that JEV ecology in Australia remains poorly understood. This study's results can support improvements in arbovirus surveillance systems, promoting earlier detection of circulating viruses. Increased focus on vector ecology and distributions is crucial for better understanding JEV transmission in Australia.

## Author summary

Our work represents a major advance in understanding the distribution of mosquito vectors and vertebrate hosts for Japanese encephalitis virus (JEV) in Australia. Prior to the summer of 2021/22 JEV was known from periodic incursions into Australia's northernmost extremes, but from December 2021 there was an unprecedented expansion into southern Australia associated with abundant breeding habitat for hosts and vectors associated with a La Niña weather pattern. We brought together mosquito occurrence records, and structured occurrence data for ardeid hosts to inform time-varying, continental scale habitat suitability models for Australian JEV vectors and vertebrate hosts. We present a method of rescaling and combining the joint model outputs to facilitate consideration of a combined model of suitability for pathogen transmission.

## Introduction

Japanese encephalitis virus (JEV) is a mosquito-borne, zoonotic virus endemic to Southeast Asia and the Indian sub-continent. The virus circulates primarily between mosquitoes, ardeid birds, and domestic pigs [1–3]. Spillover infections in humans cause Japanese encephalitis (JE) disease, which can result in serious and potentially fatal illness. It is estimated that over 100,000 cases of JE and around 10,000 deaths occur each year in endemic countries, mostly in young children [1].

Between 1995 and 2005 JEV was periodically detected in the islands of the Torres Strait off the northern tip of the Australian mainland, and sporadically detected on mainland northern Australia, but there was no evidence that the virus had become endemic despite the abundance of competent hosts and vectors [4]. In 2021, a fatal case of JE was recorded from Bathurst Island in the Northern Territory. In 2022 JEV was detected in various species across a wide geographical area that encompassed a number of locations in northern Australia and as far south as the southern states of Victoria and South Australia, a latitudinal expansion of over 24 degrees. From 2020 spring and summer rainfall over eastern Australia was influenced by a multi-year "La

Division, Department for Health and Wellbeing, Adelaide. mailto:healthprotectionprograms@sa.gov.au Tasmania: Infection Prevention and Control Unit, Department of Health, Hobart. tipcu@health.tas.gov.au Victoria: Community and Public Health Division, Department of Health, Melbourne, Victoria. mailto:infectious.diseases@dhhs.vic.gov.au Western Australia: Environmental Health Directorate, Western Australian Department of Health, East Perth. mailto:medical.entomology@health.wa.gov.au.

**Funding:** This work was funded by the Australian Commonwealth Government Department of Health and Aged Care, through a research contract awarded to FMS and NG. The funder played no role in study design, analysis, preparation of the manuscript or the decision to publish.

**Competing interests:** The authors have declared that no competing interests exist.

Niña" Southern Oscillation pattern [5]. La Niña years greatly increase the likelihood of heavy rainfall and flooding in eastern Australia and subsequent flow through inland river systems, resulting in increased habitat for mosquito larvae and ardeid birds. A total of 42 cases of JE were diagnosed in humans in Australia including seven deaths to June 2022 [6,7], as well as many more detections in pigs from commercial premises [3] and in feral pigs in the north of the country. There were more than 50 infections detected in feral pigs in the Northern Territory alone in 2022 [8]. There is concern that environmental conditions that support populations of vectors and hosts for JEV, and other flaviviruses, could become more frequent [3,9,10] (though counter-vailing influences are also possible, see [11]), underlining the importance of under-standing transmission risk for JEV in space and time.

In Southeast Asia JEV has been linked to serious disease in humans for over a century, though knowledge of the vectors and vertebrate hosts from which trans-mission may spill over to humans remains incomplete (e.g., [12–14]). JEV has been isolated from over 30 mosquito species in five genera across its distribution [15,16], but previous assessments of JEV transmission in SE Asia have identified *Culex tritaeniorhynchus* mosquitoes as the primary vector [12,15,17]. Herons and egrets are considered the fundamental wildlife host group [12–14,17,18], and swine can play an amplifying role where spillover transmission occurs [2,19]. Given considerable uncertainty remains about vectors and vertebrate hosts in SE Asia where JE routinely represents a significant health burden, the level of uncertainty is particularly high in Australia. Here JE is an emerging infectious disease [6,19] and the putative wildlife reservoir and vector species are potentially different than in the endemic range of the virus [3,4,18].

Members of the *Culex* genus are the primary vectors of JEV throughout the distribution of the virus. The dominant JEV vector in SE Asia is *Cx. tritaeniorhyn-chus* and, whilst that species has recently become established in Northern Australia [20,21], Australian field studies and laboratory-based vector competence experi-ments have incriminated *Cx. annulirostris* as the primary local vector [4,15,22]. *Cx. annulirostris* occurs widely across the Australian mainland, where it uses temporary and semi-permanent groundwater as larval habitats, including freshwater wetlands, drainage ditches associated with agriculture, small transient pools persisting after significant rainfall or inundation, as well as highly-polluted and brackish water sites [23]. The species is temperature and moisture sensitive (e.g., [24]), and can be abundant during the northern wet season and temperate summer, particularly fol-lowing heavy rainfall (e.g., [9,25,26]). While *Cx. annulirostris* is considered the most important JEV vector (and for closely related flaviviruses Murray Valley encephalitis virus (MVEV) and West Nile virus (Kunjin strain, WNV$_{KUN}$) [27]) in Australia, there is laboratory or field evidence to suggest that *Cx. bitaeniorhynchus*, *Cx. gelidus*, *Cx. palpalis*, *Cx. quinquefasciatus* and *Cx. sitiens* could in principle play a role in JEV transmission [22,28].

Herons and egrets – wading birds in the family Ardeidae (hereafter 'ardeids') – have been identified as the fundamental wildlife host species group where JEV rou-tinely circulates in Asia [12–14,17,18]. The ardeids are well represented in Australia,

with some species implicated as competent hosts in experimental studies (e.g., [29,30]). Herons and egrets are water dependent, constructing nests in trees or shrubs fringing or overhanging water. They are ubiquitous around estuaries and wetlands of the coast and hinterland and can also be abundant through Australia's vast semi-arid to arid inland areas where they track floodplains and myriad wetlands that are periodically inundated due to lateral flow from higher rainfall areas in the tropical and sub-tropical north [31–33]. Ardeid species, like other waterbirds, can track resources over large distances and thus could be important in rapid transport of JEV to new areas [34].

The potential role of pigs in amplifying JEV circulation in Australia is less clear. In 2021–2022 the virus was initially detected in feral pigs and in many commercial piggeries [3,7], though only one human case was linked to a commercial pig premises by epidemiological investigation [7]. Investigations of previous incursions of JEV into northern Australia showed serological evidence of infections in naive domestic pigs and those deployed as sentinel animals [28]. The use of pigs as sentinel animals for virus surveillance in remote northern villages was discontinued for ethical reasons, cross-reaction with endemic flaviviruses in serological assays and fear of contributing to transmission [35]. There are no native swine in Australia but geographically-limited surveys over the years have given rise to estimates of tens of millions of animals over a distribution encompassing around half of the Australian mainland [36–38], suggesting they could contribute as a free-ranging wildlife reservoir. While feral pigs may play a role in JEV transmission – and despite their recognition as a destructive pest in Australia – until very recently there had been no major update to the piecemeal datasets and models representing their distribution. Therefore, we excluded them from our conceptualisation of the wildlife reservoir in this work.

For zoonoses like JEV that periodically spill over from the virus' vertebrate host reservoir to humans, it is important to identify not only the key vector and host species but also the geospatial and temporal characteristics of climate and landscape that bring them together to generate permissive conditions for pathogen transmission [39]. Mosquitoes, ardeid birds and feral pigs are widely distributed in Australia outside the arid zone and their habitat preferences converge around fresh to brackish water sources, wet and humid environments, and warmer temperatures. Australia encompasses a large latitudinal range from tropical to temperate biomes, and over much of that range exhibits highly variable climate, so the spatial distribution of favourable conditions for hosts and vectors is likely to be commensurately variable. The task of estimating spatial foci of pathogen transmission and infection risk has been made easier in recent decades by the greater availability of spatial data at temporal and spatial scales relevant to the dynamics of sylvatic cycles [40,41]. These data facilitate the translation of heuristic understanding and limited spatio-temporal occurrence data of vectors and hosts into quantitative, spatially explicit, and testable representations of habitat suitability for pathogen transmission [42]. The most commonly applied approach is Habitat Suitability Models (HSM, also widely referred to as environmental niche models [42,43] or species distribution models [44]). HSMs typically combine high-resolution climate and land cover data with species occurrence data (such as those stored in national or global repositories such as GBIF [45]) that may be derived from various sources including citizen scientist observations or structured scientific surveys. Habitat suitability models can be considered either temporally 'static' or 'dynamic' depending on whether occurrence data are assigned covariate data contemporaneous to the observation or longer-term statistical summaries of climatic and other time-varying covariates [46,47].

In response to the significant uncertainty about JEV transmission ecology in Australia, we produced time-varying geospatial models of the relative suitability of habitat conditions for *cohorts* of purported mosquito vector and ardeid host species groups that together are likely to encompass the wildlife transmission cycle for JEV. In addition, acknowledging the weight of empirical evidence and expert opinion, we built a separate model for *Cx. annulirostris*. Furlong et al. [18] recently published the first spatial models of the relative habitat suitability in Australia for candidate JEV vector species in the *Culex* genus. That work linked available biodiversity atlas records for the selected *Culex* species to long run statistical summaries of central tendency and variation in environmental and climatic variables corresponding to those observations. For *Culex* mosquito species occurrences we combined publicly available data with a significant, newly collated national dataset of mosquito trap observations from arbovirus surveillance programs conducted by state and territory health

departments. This resulted in a model training dataset with more than 20-times more records. For ardeid hosts, we used curated checklist records from eBird and Birdlife Australia comprising quantitative information of spatially-varying detection effort. For both hosts and vectors we built habitat models from month-scaled occurrence and covariate data from 2000–2023. We rescaled the host and vector model outputs corresponding to the peak of the outbreak to map combined habitat suitability for JEV in mainland Australia and nearby islands.

## Methods

### Selection of species for a cohort model

Our model of the relative suitability of habitat for mosquito vectors was based on occurrence data for selected species in the genus *Culex* with some demonstrated potential for a role in JEV transmission namely: *Cx. annulirostris*, *Cx. bitaeniorhynchus*, *Cx. gelidus*, *Cx. palpalis*, *Cx. quinquefasciatus*, *Cx. sitiens*, and *Cx. tritaeniorhynchus* [22,28]. There is uncertainty about the vectorial capacity of these species in different Australian climate zones (see map S1 Fig), plus potential for taxonomic uncertainty (viz. *Cx. tritaeniorhynchus*, *Cx. sitiens* subgroup), so distinguishing between the distributions of these species seemed unlikely to contribute to our aim of national scale mapping of habitat suitability for JEV transmission. Therefore, occurrence data for these species (see S2 Fig) were combined to produce a cohort model; our mosquito vector habitat suitability model predicts areas where conditions are likely to be suitable for the presence of any one of these species. However, due to published weight of opinion about the primacy of *Cx. annulirostris* as a vector in Australia [15,22,28,48], we also developed a separate model restricting presences to *Cx. annulirostris*, using the same method as described below. The results are included in the Supplementary information. Given the broad similarity between the single-species model and the *Culex* cohort model, we focus our presentation on the cohort vector model in defining the potential habitat suitability for JEV transmission in Australia.

We also adopted a species cohort model approach for wildlife hosts. The target group for constructing a model of habitat suitability was narrowed to herons and egrets (Family Ardeidae): *Ardea alba* (great egret), *A. intermedia* (intermediate egret), *A. pacifica* (white necked heron), *Bubulcus ibis* (cattle egret), *Egretta garzetta* (little egret), and *Nycticorax caledonicus* (Nankeen night-heron) (see S3 Fig). These species were identified on the basis of published information and expert judgements about their: habitat associations; breeding ecology and movement patterns (e.g., [49]); viraemia; and evolutionary similarity to species implicated in JEV elsewhere in Asia through antibody prevalence or infection competence (e.g., [13]).

### Data for mosquito vector species

All data preparation, model development and implementation described below was performed in R v4.2.2 [50]. After filtering and cleaning we retained 15876 occurrence records (unique sampling locations and months) for one or more of the selected *Culex* species of interest. The majority of the records (98%) were mosquito surveillance data obtained from Australian state and territory health authorities (Table 1).

### Mosquito surveillance datasets

Arbovirus surveillance programs, including mosquito surveillance activities, are implemented independently by each state jurisdiction [35,66] with varying degrees of devolved or centralised responsibility for data management, specimen identification, and reporting. While we were able to collate mosquito trap data for each jurisdiction, the resulting dataset does not represent a comprehensive picture of Australian mosquito surveillance over space and time. We obtained records for between 1–22 years depending on the jurisdiction (Table 1), and we know for example that records from commercial premises were removed from some jurisdictions' data. Records were typically generated from carbon dioxide-baited encephalitis virus surveillance (EVS) or Centers for Disease Control (CDC) light traps, or similar. Some trap locations

**Table 1. Summary of vector occurrence datasets for the period 2000–June 2023 obtained to support habitat suitability modelling. Trap events refer to unique combinations of location (1 km raster grid cell) and year and month of sampling.**

| Source | Year range | Unique locations | Trap events |
|---|---|---|---|
| Mosquito surveillance data | | | |
| New South Wales | 2000–2023 | 279 | 5208 |
| Western Australia | 2000–2023 | 430 | 4575 |
| Victoria | 2014–2023 | 356 | 2450 |
| South Australia | 2021–2023 | 254 | 650 |
| Queensland | 2000–2023 | 374 | 503 |
| Northern Territory | 2021–2023 | 131 | 473 |
| Public databases | | | |
| Atlas of Living Australia (ALA) [51] | 2000–2023 | 146 | 181 |
| Global Biodiversity Information Facility (GBIF) [52] | 2000–2023 | 86 | 108 |
| Published literature | | | |
| van den Hurk et al 2002 [53] | 2000–2000 | 20 | 20 |
| Frances et al 2009 [54] | 2005–2006 | 5 | 19 |
| Hall-Medelin et al 2012 [55] | 2004–2006 | 8 | 8 |
| Ramirez et al 2020 [56] | 2018–2018 | 8 | 8 |
| Phillipe-Janon et al 2015 [57] | 2014–2014 | 3 | 6 |
| Ritchie et al 2007 [58] | 2003–2005 | 2 | 5 |
| Jansen et al 2009 [59] | 2008–2008 | 1 | 2 |
| Johansen et al 2003 [60] | 2001–2001 | 1 | 2 |
| Sebayang et al 2021[1] [61] | 2020–2020 | 1 | 2 |
| Jansen et al 2013 [62] | 2005–2005 | 1 | 1 |
| Johansen et al 2004 [63] | 2000–2000 | 1 | 1 |
| Johnson et al 2015 [64] | 2014–2014 | 1 | 1 |
| van den Hurk et al 2007 [65] | 2002–2002 | 1 | 1 |

[1]Trap data from tree canopy (>2 m above ground) were excluded.

were represented by a single trap event but others had been sampled regularly for many years resulting in hundreds of sampling events at the same location. Trapped mosquitoes had been identified and counted by trained entomologists. We discarded records for traps that were marked as having malfunctioned, samples that had not been resolved to species level or where species identification was not conducted, and records that could not be assigned a location with the required spatial precision of ≈1 km. For jurisdictions where an abundance value for individual species was provided, records with 0 abundance for all species of interest were categorised as a non-detection and added to the background pseudo-absence data.

## Public repositories and other published datasets

We augmented the data from jurisdictions' surveillance with occurrence data for the same period published in literature or in publicly accessible databases (Table 1). Our literature extraction effort was not uniform, focussing on remote areas in the north-eastern jurisdiction of Queensland where routine trapping is difficult and sampling was usually undertaken in a single location at a single time-point (e.g., [53,55]). Those published records were all from $CO_2$-baited light traps, except Sebayang et al, who used $CO_2$- and octenol-baited POD traps. Additionally, occurrence data for mosquito species in Australia from the observation protocols 'human observation', 'machine observation', and 'pickled specimen' were obtained

from the Atlas of Living Australia (ALA) [51] and Global Biodiversity Information Facility (GBIF) [52] (Table 1), derived from miscellaneous sampling methods. GBIF records were filtered to exclude those with geographic uncertainty of ≤ 1 km$^2$ or flagged as having potentially erroneous or unreliable spatial location attributes. ALA records were filtered to include only those flagged as appropriate for Collaborative Species Distribution Modelling. These databases include observations and data from citizens and professional scientists; subsuming the records of casual observations from iNaturalist, returns from licensed scientific collection, and other specific projects and databases auspiced by biodiversity and ecological monitoring programs of Australian states and territories.

## Data for ardeid host species

Bird observation checklists were obtained from eBird [67] and Birdlife Australia [68] for the period January 2000–June 2023. Checklist submissions to both datasets are organised using unique identifiers attributable to an observer at a given location at a certain time (though from Birdlife Australia we could only access the observer identifier at the level of the data custodian). Contributors can mark checklists they submit as "complete", warranting that they recorded all and only species that were seen or heard during an observation event. The observation events are defined by metadata tags indicating the observation protocol (e.g., [69]) and search effort (time, distance, etc). "Complete" checklists with suitable search effort attributes can therefore be used to model data as presences (detected) and absences (zero-filled records of non-detections) (e.g., [70]). We used the locations of complete checklists with detections of any of the selected ardeid species (presences) and the locations of all other complete checklists (absences) to generate the habitat suitability model for ardeid hosts (see Table 2).

On examination of the data, it was noted that some individual observers had submitted many checklists for a given location within a short time-interval. Other observers may also have independently or knowingly submitted checklists for a given location in the same period. Therefore, the raw dataset contains many observations that are pseudo-replicates for our purposes. Spatial and temporal thinning of records from species occurrence databases has been recommended to ameliorate the biases that characterise such unstructured data [70,71]. However, these characteristics of the data were common to presence and absence checklists and therefore do not constitute an issue of imbalance in spatial sampling

**Table 2. Summary of heron and egret occurrence data from selected field observation protocols obtained from Birdlife Australia (BA) and eBird (eB) for the period 2000–June 2023. The counts are unique, complete checklists ("lists") recording detection of the target species ("present"), any one of the target species ("Any ardeid"), or asserting absence of all. Overall empirical prevalence ("prev.") is also reported. To form the model dataset, for any cell in any month with records from across the included protocols, a single observation representing presence or absence was randomly drawn in proportion to prevalence per cell per month.**

| | Observation protocols | | | | | | | |
| --- | --- | --- | --- | --- | --- | --- | --- | --- |
| | 500m area search | | Stationary | | Traveling* | | 2ha-20min search‡ | |
| | lists | prev. | lists | prev. | lists | prev. | lists | prev. |
| *Ardea alba* | 9579 | 0.06 | 7500 | 0.06 | 16876 | 0.08 | 5946 | 0.02 |
| *Ardea intermedia* | 3158 | 0.02 | 3397 | 0.03 | 7918 | 0.04 | 2246 | 0.01 |
| *Ardea pacifica* | 3806 | 0.02 | 2600 | 0.02 | 4924 | 0.02 | 2874 | 0.01 |
| *Bubulcus ibis* | 3809 | 0.02 | 4722 | 0.04 | 8766 | 0.04 | 3051 | 0.01 |
| *Egretta garzetta* | 3742 | 0.02 | 3270 | 0.03 | 6895 | 0.03 | 1819 | 0.01 |
| *Nycticorax caledonicus* | 1446 | 0.01 | 853 | 0.01 | 3137 | 0.01 | 1075 | 0.00 |
| Any ardeid | 18005 | 0.11 | 15161 | 0.12 | 34306 | 0.16 | 13157 | 0.04 |
| Total | 161538 | | 121663 | | 216525 | | 321253 | |

* Filtered to include only records with a recorded survey distance < 1 KM.

‡ This protocol was excluded from model due to markedly different empirical prevalence at ardeid cohort level.

bias so much as spatio-temporal pseudo replication with respect to the resolution of our covariates (1 km grid cell with monthly averages). For a given 1 km grid cell, per month, we randomly selected a single observation of presence or absence in proportion to the empirical prevalence from observations made during daylight hours in that cell and month. We combined data from several observation protocols as equivalent on the basis of similar empirical prevalence (see Table 2) and overlap in definitions; in each case the submitter warrants that the primary purpose of activity generating the record was bird observation, and supplies the start time, duration, location and distance/area covered [72,73]. A "500m area search" is a Birdlife Australia protocol defined as a search with a maximum displacement of 500 m from the recorded starting location. A "stationary" search is eBird's equivalent protocol, but with a 30 m limit on displacement. eBird's "travelling" protocol is for activities with greater than 30 m displacement, though the trajectory and distance must be recorded. We excluded records where a search area was likely to have encompassed more than 1 km$^2$ in extent. The "2ha-20min search" is an Australian research protocol found in both databases that imposes a fixed effort in space and time. Preliminary analyses indicated that there was a reduced prevalence of the target species associated with the 2 hectare-20 minute protocol. We could not accommodate an effect of protocol in our geospatial model, so records obtained via that protocol were omitted from the model dataset.

## 'Static' land cover variables

We constructed 1 km grids summarising the fractional cover of eight 'static' land cover types denoting vegetation structure and with special emphasis on water features. *Culex* mosquitoes and ardeids are strongly associated with the availability of still or slow-moving surface water such as naturally provided by wetlands, estuaries, tidal flats and billabongs created by receding flow through anabranching watercourses. These associations can be complex; for example habitat suitability for some ardeids in the Northern Territory seems higher in the drying phase of the tropical wet [74]. Expansive artificial habitats can be important too, as seen in the increase of *Culex* habitat due to irrigated agriculture in Asia [75]. This process has its analogue in parts of Australia's interior where cotton and rice farming has expanded, and other irrigation schemes with similar water demands [76]. *Culex* species, including *C. annulirostris* can also reach high abundance in freshwater habitats in urban and peri-urban contexts particularly those subject to eutrophication (e.g., [77]), whilst small artificial habitats such as containers can favour abundance of species like *Cx. quinquefasciatus* [78]. Though productive habitats can exist within urban and peri-urban areas we omitted the 'built up' land cover type from the model covariates since that category conflated urban areas with, for example, sealed roads and other surfaces in remote areas. To further expand on the representation of water features in the environment we included a layer summarising the long-term relative frequency of water observation for each pixel for the period 1987–2021 which captures areas in semi-arid zones with low precipitation that may periodically become and remain inundated under certain circumstances (Table 3).

## Temporally dynamic environmental variables

Each occurrence observation was assigned the relevant mean day- and night-time land surface temperatures; indicators of greenness of vegetation, soil moisture and soil texture; and interpolated total rainfall pertaining to the month of the observation, and the prior month. These time-varying datasets allow relative habitat suitability to be estimated as variable at a given location through time, as a function of changing resource availability, phenomena known to influence occurrence, abundance, and breeding success of ardeid hosts and mosquitoes (e.g., [86,87]). Global correlation among the covariates is presented in S4 Fig and the spatial pattern is shown for selected covariates in S5 Fig.

## Strategy to minimise sampling and reporting bias

The sample or observer bias inherent in occurrence datasets used in habitat suitability modelling pose problems for their intended use [88], though some advances can mitigate these [89,90]. Since our objective was to produce continental scale models of relative host and vector habitat suitability for mainland Australia and nearby islands, the optimal datasets

**Table 3. Metadata for spatial covariates.**

| Variable | Definition | Source |
|---|---|---|
| 'Static' fractional land cover (0–1) | | |
| Frequency of inundation | Base 30 m product represents for each pixel the proportion of all cloud-free images in which it was deemed occupied by water for the period 1987–2021. | GeoSciences Australia 'Water Observations from Space' [79]. The original product was summarised to average frequency calculated for 1 km grid. |
| Permanent water | Area covered for ≥9 months of year by fresh or salt water. | European Space Agency [80]. The original 10 m categorical product (see Table 2 of [81]) was aggregated to proportional cover of each type 1 km grid. |
| Herbaceous wetland | Natural herbaceous vegetation permanently or regularly flooded. | |
| Mangrove | Intertidal zones dominated by treed vegetation. | |
| Grassland | Area dominated by herbaceous vegetation of ≥ 10% cover, may include uncultivated pasture or crop. | |
| Cropland | Occupied by annual crops harvested at least once within 12 months of sowing. | |
| Bare / sparse | Exposed substrate, always less than 10% vegetated cover. | |
| Shrubland | Natural shrub cover of ≥ 10% or more. | |
| Forest | Tree cover of ≥ 10% cover. Plantations, swamp-forest captured in this class and canopy may cover other types (e.g., water bodies). | |
| Temporally dynamic environmental variables | | |
| Night temperature (Night T °C) Day temperature (Day T °C) | Average Land-surface temperatures derived from MODIS v6 8-daily 1 km product MOD11A2 | MODIS base products were post-processed by a global implementation of Weiss et al [82] algorithm to gap-fill cloud-affected scenes. The gap-filled outputs were aggregated to a monthly mean. |
| EVI (Greenness) | Enhanced Vegetation Index - composite monthly product robustly emphasising vegetation canopy density and greenness against soil and structure background variation (see MOD13A2 [83]) | Regenerated from raw MODIS products MCD43D62–MCD43D68 and post-processed by a global implementation of Weiss et al [82] algorithm to gap-fill scenes affected by cloud. The gap-filled outputs were aggregated temporally to a monthly mean |
| TCW (Wetness) | Tasselled Cap Wetness - Reductive transformation and rotation of Landsat TM reflectance bands emphasising moisture variation in soils and possibly vegetation [84] | |
| TCB (Brightness) | Tasselled Cap Brightness - Weighted sum of Landsat TM reflectance bands emphasising variation in soil physical properties [84] | |
| Rainfall (mm) | Monthly rainfall total (mm) according to the interpolation algorithm of Climate Hazards group Infrared Precipitation with Stations (CHIRPS) [85]. | Resampled to 1 km grid. |
| Rainfall (lag 1 month) | Monthly rainfall total (mm) as above from CHIRPS for the month preceding the observation. | |

would result from random survey for our species of interest (mosquito trap deployment or bird observation protocol enacted), stratified across covariate space – in our case that implies repetition across months and over the years – with both presence and absence outcomes being recorded with equal probability. Such statistically ideal datasets do not exist and they would be expensive to generate. Both ardeid and mosquito datasets were spatially biased, but due to the different characteristics of the datasets, bias mitigation required different approaches.

The mosquito surveillance trap networks that generated around 98% of our vector dataset are operated or commissioned by Australian State and Territory health authorities. They were originally designed to detect pathogens of zoonoses of public health concern, primarily Murray Valley encephalitis virus (MVEV), Ross River virus and Barmah Forest virus. These trap locations are spatially biased in the sense that the majority were purposefully deployed close to likely mosquito breeding habitat (wetlands, drainage features, water holding ponds) near resident human populations of town and peri-urban areas, or other areas deemed to have elevated exposure risk. A consequence of this understandable bias is that the great majority of sampling locations were constrained to a subset of habitat types for any given latitude, though coastal and inland conditions represent considerable amplitude in their own right, e.g., [91]. Although these trap datasets included events where none of the species of interest were recorded ('absences'), we opted to combine the trap detections with other occurrence records and treat the dataset as occurrence (presence-only) data.

There are three main options for mitigating spatial bias in occurrence (presence-only) data. 1) Include a covariate that offers a good representation of the anticipated bias in the model training, and replace that covariate in predictions with a fixed value to standardise the bias over the prediction area, e.g., [89,92]. 2) Construct a statistical model of the observation process that is presumed to generate the bias, e.g., [93]. These options complement the use of randomly selected background points, though the latter also requires at least some true presence-absence data to inform the observation model. A third option, 3) replaces random background points with target-group background points whose own generative process is presumed to have a similar bias as the occurrences, e.g., [90]. The target-group background, in reflecting the spatial (and environmental) bias present in the occurrence data, enables a prediction that is less biased towards areas over-represented in the occurrence set. We adopted the target-group background approach and fitted a model that contrasts the locations of mosquito occurrences against the locations of a target-group background dataset.

To mitigate spatial reporting bias with the target-group method we required a target group for whom observations would have been generated via a similar sampling methodology, but not sharing all similar environmental habitat requirements with the model target, otherwise the model will fail to capture those shared environmental characteristics in predictions of the species of interest. For example, if we had selected records for Culicidae, or even Insecta as the target group, it may have removed the effects of temperature and humidity from the model (since these are common environmental requirements for insect life). The resulting predictions would be biased away from these areas. Instead we obtained species occurrence records in Kingdom Animalia from GBIF [94] to use as our target group. Animalia occupy all habitats but the pattern of observation and reporting of those organisms is expected to be biased toward population centres and easily accessible areas [95,96]. By selecting records of Animalia, we therefore aimed to capture general patterns of bias in population density, accessibility and biological data recording in Australia. We arbitrarily limited the size of the Animalia background dataset to the same size as the mosquito occurrence set, and supplemented those with the mosquito trap records with zero abundance for the species of interest.

Our mosquito occurrence dataset also reflected temporal bias. Compared with the Animalia target group dataset, which suggests a steady accumulation rate through time, a quarter of the vector surveillance records we were able to compile date from after 2020 (see S6 Fig). This may be due in part to trapping having increased through time, particularly in response to new outbreaks, but primarily reflects that the data supplied to us by some state and territory authorities was limited to the period from 2021 onwards. We addressed this bias at the level of state and territory by sampling target-group background records only in years for which we had occurrence data for that state or territory.

Because we interpreted the curated datasets from eBird and BirdLife Australia's as equivalent to presence-absence survey data, the treatment of spatial and temporal bias was simplified. In effect, we assumed that observers were equally as likely to record a 'complete checklist' in sites where our ardeid cohort occurred as in other locations, over space and through time. Consequently, whatever spatial reporting bias that applied to the presences would have been shared by the absences. Therefore, other than thinning of records described earlier, we modelled the distributions of the ardeid cohort without further accounting for spatial bias.

## Modelling approach

**Penalised logistic regression for relative abundance of *Culex* mosquito species cohort.** Relative habitat suitability for *Culex* species was modelled as a penalised logistic regression allowing for linear, hinge, or quadratic univariate effects and linear interactions between variables using the maxnet [97] and ENMeval R packages [98]. The model of the relationship between the vector of presences and background points (observations of organisms in Animalia with the selected *Culex* presences added to them) and the environmental predictors can be considered as a thinned inhomogeneous point process, where the occurrence data are interpreted as spatially dimensionless points (the observation has no assumed search area dimension) with environmental attributes [92,99,100]. As we aimed to neutralise sampling bias by the use of similarly biased target-group background points, the model response was expected to be proportional to the intensity of occurrences of at least one of the selected *Culex* species per unit area. Given the expectation of significant spatial bias in training data and model run times of several hours we benefited from the convenience functions of ENMeval for batch processing of various implementation settings for feature classes and regularization multipliers. We preferenced simpler models with more constrained features (higher regularization multipliers of 3–5) that had similarly high ability to correctly separate binary outcomes in training and testing datasets (folds), and smaller difference between theoretical and observed omission (false negative) rates at 0 and 10% thresholds, following [71]. The model presented below used a regularization multiplier of 3. Predictions of vector habitat suitability were made to each month from July 2021–June 2023, and selected previous Summer (February 2019, 2020) months. We report omission rate statistics and a continuous Boyce index [101] (CBI), which tests whether areas of increasingly higher predicted suitability coincide with higher density of occurrence points. CBI values approaching 1 indicate that the values predicted by the model are consistent with the distribution of occurrence points.

**Boosted Regression Tree of probability of ardeid host species occurrence.** Relative habitat suitability for ardeids was modelled using a gradient-boosted regression tree (BRT) modelling framework, fit with the gbm3 v3 R package [102]. BRTs combine desirable elements of regression and machine learning approaches [103] and have been shown to perform well in ensemble modelling of habitat suitability (e.g., [104]). Each observation checklist was considered as an independent observation event resulting in detection (seen or heard) or non-detection (neither seen nor heard), subject to Bernoulli error. The same spatial cross-validation approach was applied to the ardeid host data as described above for *Culex* data. Following [103] we first used a relatively high learning rate (shrinkage parameter) to obtain a preliminary model whose optimum cross validation performance required more than 1,000 trees. To obtain a final model, we then re-ran the model for the estimated optimal number of trees with a reduced learning rate. After filtering, our dataset comprised 20443 detections and 144084 non-detections, or an overall empirical prevalence of about 15%. With a strongly imbalanced response class, good prediction accuracy may be obtained trivially by a model that always predicts to the majority class. We trialled 'half-weighting' the occurrence data (*weight* = $\frac{1}{N_{class}}$) to balance the contribution of classification errors in the minority class with those of the majority class in subsequent model steps. Whereas unweighted models achieved less than 20% sensitivity in internal cross-validation (and greater than 90% specificity), weighting the presences resulted in models with more than 80% sensitivity (at the expense of specificity ≈ 60%). Covariate interaction depths of 2–6 were trialled, with comparisons of cross-validation confusion matrices usually supporting the restrained model complexity of up to 4. We set the bag fraction to 0.75, enforced balanced class representation in cross-validation folds, and set a minimum of 10 observations as the terminal node size as a limit on overfitting. We report the area under the Receiver Operating Curve (AUROC) from a holdout set of 10% of data as a measure of prediction accuracy. Additionally, we visually compared prediction scenes with independent waterbird observation datasets consisting of landscape scale observations from light aircraft [105]. As with the vector model, we predicted ardeid habitat suitability to all months from July 2021–June 2023.

**Spatial cross-validation.** For both vector and host models observations were assigned to spatial blocks for cross-validation (see S7 Fig), which is a method employed when prediction beyond the geographic range of the occurrence data

is an objective [106]. The maxnet and BRT approaches employ cross-validation folds slightly differently. In maxnet models, the resulting ensemble predictions from testing folds are evaluated against the data in training folds after estimation as a measure of the model's ability to correctly predict out-of-sample data. In BRTs a similar spatial cross validation process occurs with each tree added during the model fitting procedure, constraining variable selection, model complexity and other parameters.

The spatial cross-validation tile grid was generated using the R package blockCV [107], which provides a function to estimate an appropriate resolution for the grid considering spatial autocorrelation in covariate rasters. A grid of tiles approximately 230 km was applied and observations and background points were assigned to the corresponding tile by location. For hosts and vectors each tile's data were then randomly assigned to one of five cross-validation folds to obtain approximately even numbers of presence and absence/background points per fold.

### A combined model of potential JEV vectors and vertebrate hosts

BRT and maxnet algorithms have both been shown to yield similarly accurate predictions for common datasets (e.g., [92,99]), but our host and vector habitat suitability models return metrics of suitability that are not directly comparable. The presence-background modelling of *Culex* species estimates relative abundance of species in the cohort (after accounting for spatial bias in reporting), whilst the presence/absence modelling of Ardeids provides an estimate of the probability of species occurrence. Both vector and host suitability metrics would be expected to be non-linearly, positively related to suitability for the pathogen, which depends on the abundance of both vectors and hosts. Therefore, using predictions for February 2022– the month before the declaration of a Communicable Disease Incident of National Significance when detections were simultaneously emerging across eastern Australian jurisdictions [108], we produced a combined niche model after rescaling each output by association with publicly available data on confirmed JEV detection locations [3,109]. We first defined an empirical cumulative distribution function for hosts and vectors, respectively, based on the extracted point suitability values (for the prediction to February 2022) coinciding with the location of each confirmed JEV detection in humans or commercial pig premises. The resulting values have a common interpretation related to the proportion of detected JEV infections associated with that value or lower (e.g., a rescaled suitability value of 0.75 indicates that 75% of JEV detections were associated with the corresponding level of habitat suitability or lower, and conversely, 25% of the detections were associated with that suitability value or higher). These cumulative distribution functions for vectors and for hosts impose a common conceptual mapping between the arbitrary vector/host suitability indices and the vector/host suitability for JEV transmission (specifically the intensity of detections). We combined them into a composite suitability layer based on the minimum value of the rescaled host and vector predictors for each pixel, with the result being an estimate of the joint distribution of suitable habitat for both vectors and vertebrate hosts.

## Results

### Relative habitat suitability for vectors

Mosquito trap and occurrence data for the target *Culex* mosquito species were obtained from Australia's northern extreme in the Torres Strait Islands down to Flinders Island and Tasmania south of the mainland (Table 1, Figs 1A and S1 Fig). The surveillance trap locations that comprised 98% of our occurrence dataset were biased toward populated areas and major cities (see also S8 Fig), and in water-intensive agricultural areas, with relatively little penetration into the semi-arid and arid interior. The Animalia target-group occurrence (Fig 1B) also reflect population centres, but also tracked transportation corridors and seasonally popular tourism areas. The distribution of travel-time accessibility values for our Animalia target group background and the *Culex* surveillance records were comparable, and distinctly less biased toward major population centres than the *Culex* records obtained from public databases (see S8 Fig). Compared with the target background, there were relatively higher densities of mosquito occurrence data near remote populations in the north and north-west of the mainland, reflecting targeted mosquito surveillance priorities. None of

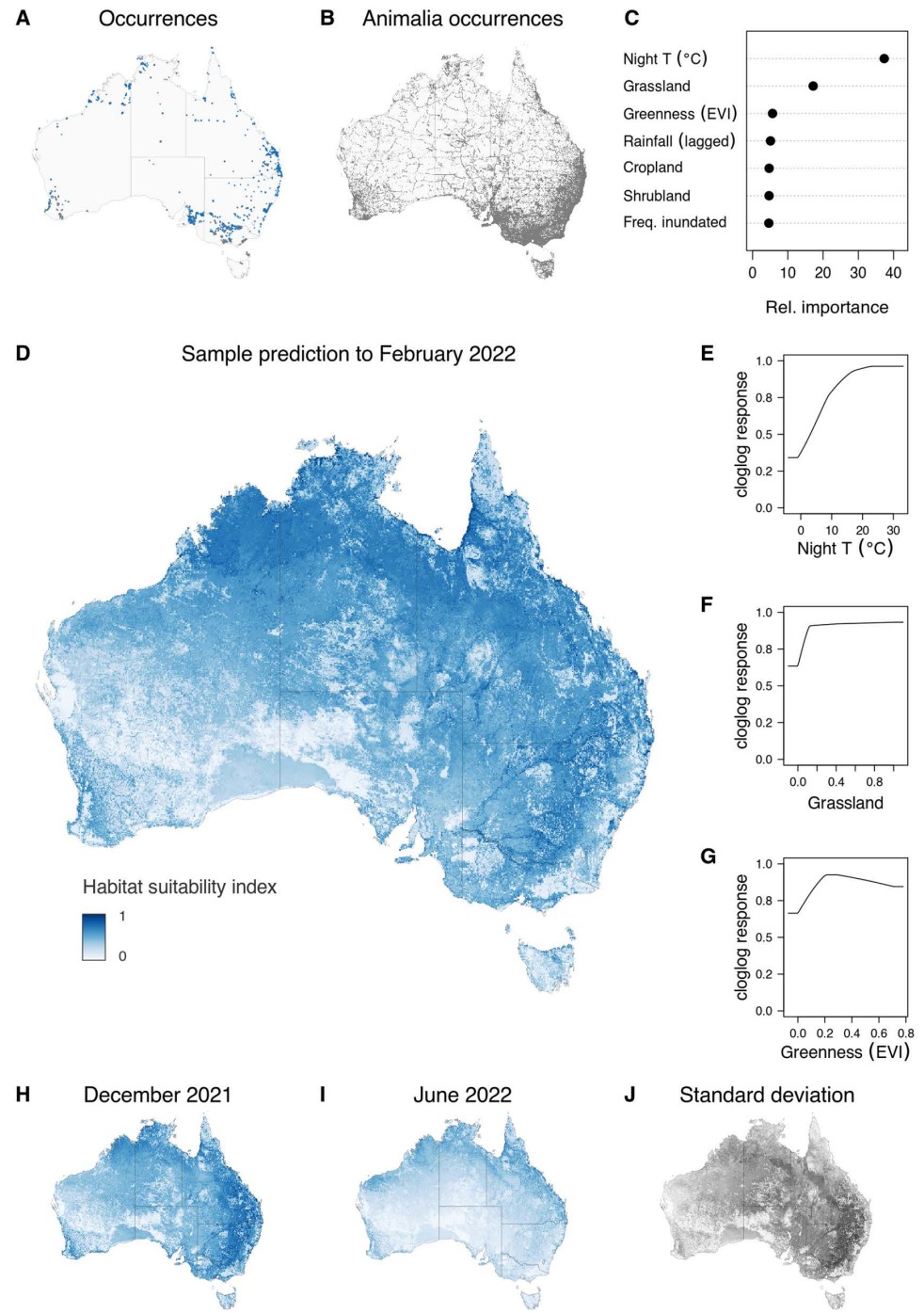

**Fig 1. Overview of input occurrence data and penalised logistic regression model of relative habitat suitability for selected *Culex* species in Australia.** A) the distribution of the combined occurrence data for *Culex* spp; B) the distribution of background occurrence data for Kingdom Animalia; C) the relative 'importance' of top seven covariates as calculated by permutation on ensemble fit; D) prediction of relative habitat suitability to the outbreak period (February 2022); E–G) the shape of the marginal relationship between relative habitat suitability and the three most influential predictor variables; H–J) prediction of relative habitat suitability from December 2021 and June 2022, and standard deviation of predictions across the temporal sequence from July 2021 to June 2023. Base maps for A, B and H–J obtained from Australian Bureau of Statistics (CC BY 4.0) https://www.abs.gov.au/statistics/standards/australian-statistical-geography-standard-asgs-edition-3/jul2021-jun2026/access-and-downloads/digital-boundary-files.

the *Culex* cohort have ever been recorded south of the mainland, including from traps recently established in Tasmania and Flinders Island in 2023 (Fig 1A). The model of habitat suitability for selected *Culex* species was built on 15539 vector occurrences and 16239 background points, including trap absences (Fig 1A and 1B). *Cx annulirostris* was the most commonly detected species, present in 90% of all cohort occurrences in the model training data. The next most common was *Cx quinquefasciatus*, which was detected in 47% of all cohort occurrences in the training data, including 85% of the occurrences where *Cx annulirostris* was not detected.

The variable most strongly associated with the pattern of habitat suitability for vectors was night-time land surface temperatures (Fig 1C). The interval of 5–15 C marked a steep increase in predicted suitability (Fig 1E). Higher grassland cover and lower-to-moderate forest and woodland cover and were associated with higher habitat suitability, though were influential over a much smaller range in the response scale (Fig 1F and 1G respectively).

The resulting spatial expression of predictions from the habitat suitability model for the selected *Culex* species was high suitability over large swathes of mainland Australia during the austral Summer of 2021/22 (December–February, Fig 1D and 1H, and see the full sequence of monthly predictions (S1 Animation in S1 File). In December 2021 high habitat suitability was predicted to extend over much of Eastern Australia except areas of high altitude and the southern island of Tasmania. Very high suitability was predicted inland of the Great Dividing Range through sub-tropical and temperate latitudes, with much more localised areas of high suitability along eastern and south western coastal margins. The pattern for February of 2022 at the peak of the outbreak was more granular in comparison. Areas of very high suitability still mapped to the area inland of the Great Dividing Range but they were less expansive than the preceding December, tracing major river systems and wetland complexes. Higher suitability was predicted for the expansive monsoon-fed savannas in the Gulf Country, and across toward the Kimberley region in tropical northwestern Australia in a broad band. Equally high habitat suitability, though less expansive in area, was predicted for long stretches of the coast across Australia's north. Extensive woodland and forest areas of tropical Queensland and north-eastern areas of the Northern Territory, which are at least seasonally wet, were predicted to have relatively low suitability. With the notable exceptions of the inland river systems in Australia's southeast, and Greater Western Sydney, the zones of high predicted habitat suitability have relatively low human population density. While some areas of highest suitability corresponded with well-sampled arbovirus surveillance locations, for others, such as the "Channel Country" around the intersection of SA, NT, QLD and NSW, there were very few *Culex* occurrence records in our dataset. Areas of higher elevation or extreme aridity stand out as unfavourable even during the outbreak period. The results of our *Culex* cohort model and those for *Cx. annulirostris* alone showed high congruence (compare Figs 1 and S9).

By winter (June 2022) suitability was restricted to areas at the tropical and equatorial latitudes (Fig 1H and 1I) due to the strong influence of night-time temperatures. Variability in predictions across the full temporal sequence (S1 Animation in S1 File) was most pronounced at sub-tropical and temperate latitudes (Fig 1J).

Post-hoc evaluation of temporal pattern in habitat suitability predictions to mosquito occurrence locations (S10 Fig) shows that the seasonality in predictions corresponds well with gross pattern in *Culex* available species abundance data. The correspondence was better in temperate and sub-tropical latitudes than in the tropical north. The vector habitat suitability model had an average AUROC on validation folds of 0.84 (range 0.77–0.89), indicating excellent predictive performance, and mean continuous Boyce index of 0.99 (range 0.98–1). The omission rate for the minimum training presence was 0 in all but one training fold, and for the 10th percentile of training presences the omission rate of the model was 0.14 on average (sd 0.08, range 0.08–0.27).

### Relative habitat suitability for hosts

Terrestrial occurrence records for the selected ardeid host species were concentrated in accessible and highly populated areas in South and Eastern Australia (Fig 2A and 2B) though at least some presence and absence records were available

PLOS Neglected Tropical Diseases

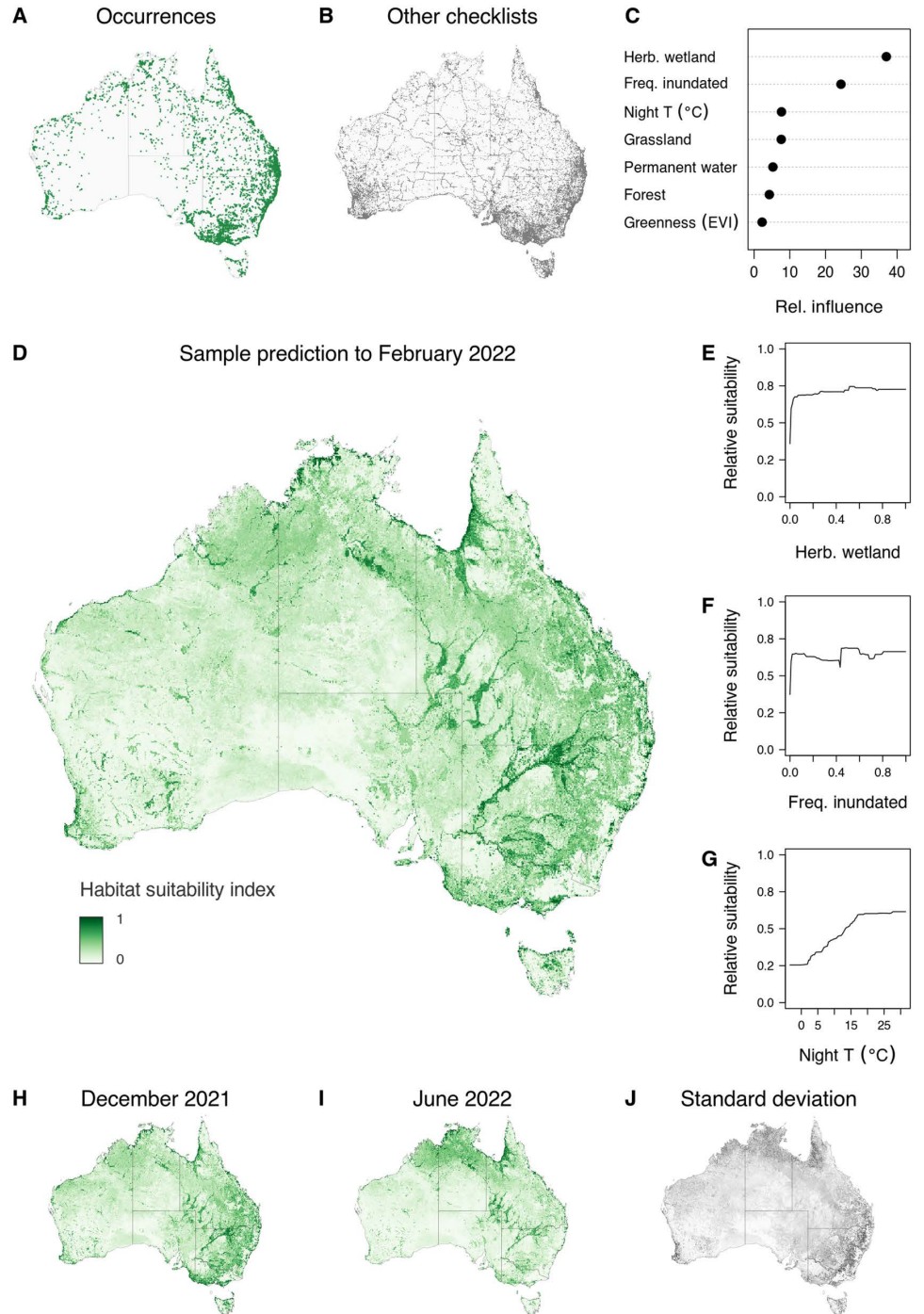

**Fig 2. Overview of input occurrence data and BRT model of relative habitat suitability for selected ardeid host species in Australia.** A) the distribution of filtered checklists for the period 2000–2023 where the selected species were confirmed sighted or heard; B) the corresponding distribution of comprehensive checklists where those species were neither seen nor heard; C) a summary of the relative influence of the top seven predictor variables submitted to the model; D) prediction of relative habitat suitability to the outbreak period (February 2022); E–G) the shape of the marginal relationship between relative habitat suitability and the three most influential predictor variables; H–J) prediction of relative habitat suitability to December 2021 and June 2022, and standard deviation of predictions across the temporal sequence from July 2021 to June 2023. Base maps for A, B, and H–J obtained from Australian Bureau of Statistics (CC BY 4.0) https://www.abs.gov.au/statistics/standards/australian-statistical-geography-standard-asgs-edition-3/jul2021-jun2026/access-and-downloads/digital-boundary-files.

over the majority of the country. Presence records were abundant in the regions around Australian population centres and along the accessible coastline, and sporadically following established transport networks throughout all but the arid zone (Fig 2A). The distribution of absence records was largely similar to the presence records, though absence records fill some gaps in the presence data that are associated with large areas of intact woodland and forest in Australia's alpine zone and semi-arid parks and reserves. Absence records also effectively trace the few transport corridors that traverse the arid zone (Fig 2B).

The bias evident in sampling intensity around population centres, being common to both presences and absences, did not strongly influence the predicted habitat suitability over the entire country. The predicted habitat suitability for ardeid hosts in February 2022 based on the model was broadly characterised by a strong coastal and hinterland emphasis on the one hand, most evident around bays and estuaries around much of Australia's coast, and the major inland drainage basins and wetland complexes of the Murray and Darling Rivers (Fig 2D). The distribution of suitable habitats comparing three sample months from 2021–2022 differs more in intensity than spatial pattern, with the inland areas predicted to be less suitable in winter (June 2022) compared with broad, enhanced suitability across northern Australia. The greatest variation over the sequence of monthly predictions was areas immediately inland from the Great Dividing Range and across parts of northern Australia (Fig 2J, and see the full sequence of monthly predictions at S2 Animation in S2 File.

The strongest covariate influences were fractional cover of herbaceous wetlands, areas that over the long term have tended to support surface water (as distinct from those with high proportional cover of "permanent water"), and mean night time surface temperatures, with a clear threshold around 10–15 C over which habitat suitability improves (Fig 2C).

The BRT model of habitat suitability for ardeid hosts had an AUROC of 0.84 when predicting to the 10% of observations (2406 presences, 15839 absences) held out from model building, indicating very good predictive performance. The model also appears to be a credible comparison against the accumulated aerial survey records of occurrence and abundance for ardeid species in the Eastern Australian Waterbird Survey (EAWS) [105], which were not used to inform our model (S11 Fig). Locations where high counts of ardeid species have been recorded in the EAWS survey corresponded to areas of higher habitat suitability predicted by our model.

### Combined model: relative habitat suitability for JEV vectors and vertebrate hosts

The model of the combined suitability for vectors and vertebrate hosts for February 2022 (Fig 3) takes the minimum value of ardeid host and vector habitat suitability at each pixel for the respective predictions to that month. At the continental scale it reflects a combination of the breadth of the habitat suitability across land cover types from vectors and concentration of the highest intensity areas in the vicinity of permanent and periodically wet drainage features more attributable to the ardeid habitat suitability model. These combined influences highlight small areas of high suitability in coastal areas across Northern Australia, and broader areas of moderately high suitability in an arc across the tropical north and down on the inland side of the Great Dividing Range, particularly near the major river courses.

### Discussion

The models of the distribution of suitable habitat for JEV vectors and hosts presented here offer the most comprehensive and temporally-dynamic representation of spatial transmission risk to date, balanced with conservatism appropriate for what is effectively a novel pathogen ecosystem over most of Australia. We sought to maximise the amount of informative data on distributions of both putative vectors and vertebrate hosts; 1) by compiling a new national dataset of mosquito surveillance trap data from almost all Australian States and Territories that resulted in a dataset 20-times greater than the mosquito records available in the public domain; 2) by accessing the substantial structured observation datasets for ardeid hosts maintained by eBird and Birdlife Australia; and 3) while taking a species cohort approach to both hosts and vectors in recognition of uncertainty about which species contribute most to JEV transmission. Importantly for a virus like

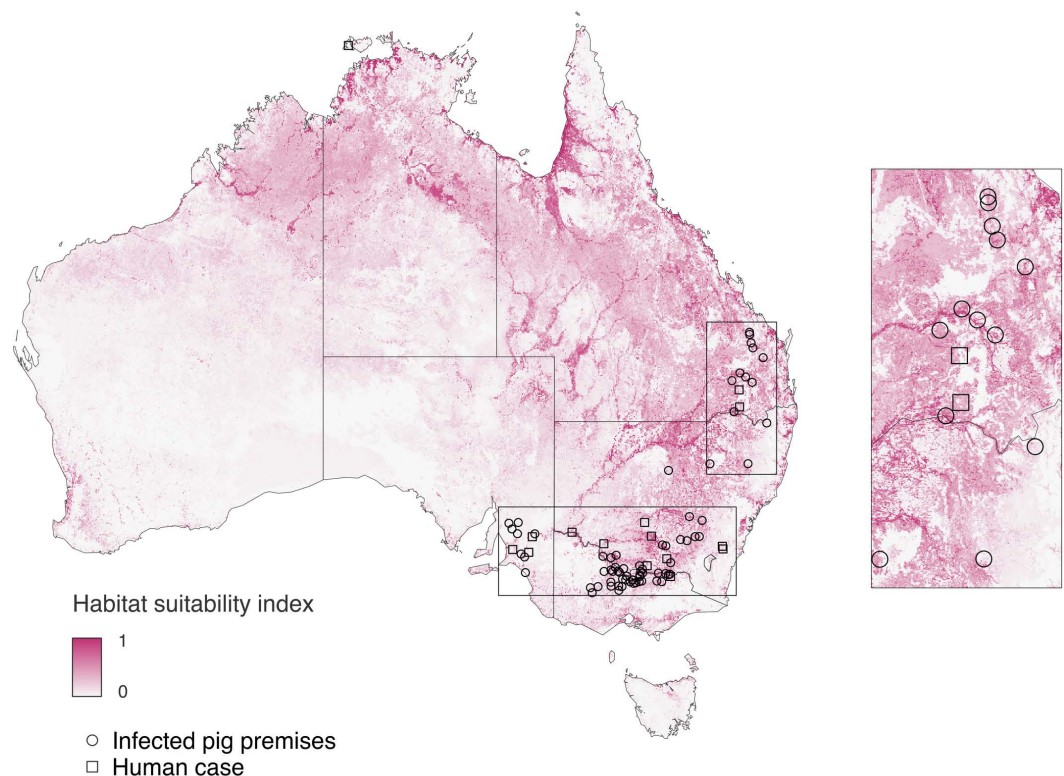

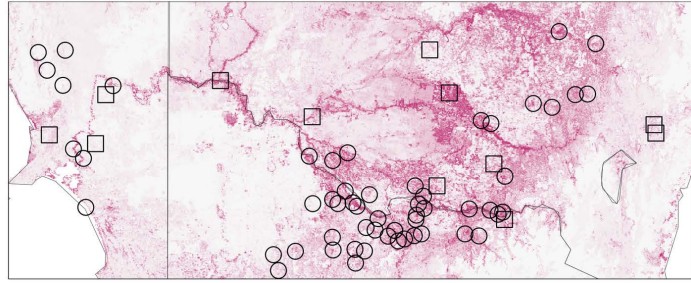

**Fig 3. Prediction of combined habitat suitability for *Culex* vectors and ardeid hosts of JEV (joint minimum rescaled suitability) for Australia to February 2022.** Point locations indicating likely infection location for confirmed detections of JEV in human cases (squares) and commercial pig premises (circles) published in the public domain to August 31, 2022 [109] used to rescale the original model outputs are over-plotted and detailed in panels. Base maps obtained from Australian Bureau of Statistics (CC BY 4.0) https://www.abs.gov.au/statistics/standards/australian-statistical-geography-standard-asgs-edition-3/jul2021-jun2026/access-and-downloads/digital-boundary-files.

JEV with a highly variable distribution in Australia, our models were built with environmental covariates for the month of each observation, allowing for predictions that express variable suitability at any given location over time.

The predicted suitability for JEV vectors and ardeid hosts was high across much of tropical northern Australia through the latter part of 2021 when infections began to be detected, and following the prodigious rainfall in northern inland areas associated with La Niña weather pattern [5]. From December 2021–February 2022 immediately prior to the period of greatest reported infections, our model predicted that those favourable conditions extended over a broad arc of eastern Australia inland from the Great Dividing Range. The combined suitability for JEV transmission was limited in broad

continental terms by the minimum night time land surface temperatures that dominated the model of suitability for vectors and then constrained geographically by the strong proximity of ardeid host habitat to wetlands and areas subject to inundation. Expansive ephemeral landscapes matching these criteria can extend south through the major drainage basins inland of the Great Dividing Range in Australia's east in La Niña years, and particularly in multi-year sequences which can have a compounding effect [5]. These are predominantly landscapes of extensive dryland and irrigated agriculture with low human population densities, which would have helped to limit the number of people at risk of exposure.

Ardeid habitat suitability was most strongly influenced by the cover of herbaceous wetland and areas frequently (though not permanently) subject to inundation where standing water becomes available. Such landscapes are exemplified by major wetland complexes and shallow meandering drainage systems of the Murray-Darling and Lake Eyre drainage basins, though the former is becoming increasingly regulated. Though our model was trained on binary presence or absence data it is broadly congruent with studies on waterbird abundance and nesting success in waterbird groups including the ardeids [110,111]. The habitat suitability models indicate low habitat suitability along the primary mountain ranges which have been shown to focus movement of GPS-tracked ibis and spoonbills [112], but see [113,114]. Further development of habitat models using ardeid abundance data may be worthwhile as areas that support large aggregations and nesting could be particularly important for JEV transmission and spread [34]. That hypothesis would be difficult to investigate, however, as JEV infections are asymptomatic in ardeids, necessitating serological and virological surveys of bird populations to assess infection status.

Night time temperature was the most important covariate in the vector suitability models, and the most influential dynamic covariate in the models of ardeid habitat suitability, with temperatures below around 10–15 C associated with markedly lower suitability. This dominance of temperature is consistent with experimental and mechanistic studies demonstrating thermal constraints on *Culex* survival and population growth parameters (e.g., [24,115,116]), and with habitat models for *Cx. tritaeniorhynchus* in Southeast Asia that also emphasised temperature as a key limiting factor [12]. By comparison, precipitation variables had little influence, despite the clear role for La Niña driven rainfall in creating ideal conditions for proliferation of mosquito and ardeid bird populations during the 2021–2022 outbreak. This may reflect the spatially and temporally diffuse relationship between rainfall and surface water availability in Australia's variable hydrological landscapes where accumulation and inundation in a given location often depends on distant rather than local catchment rainfall. We included satellite-derived variables representing inundation frequency, wetness and vegetation greenness that might capture these indirect pathways but, with the exception of long-term inundation frequency for ardeids, these too were secondary to temperature. Other studies have reported more direct precipitation effects on vector abundance [9,117] and waterbird population dynamics [33], suggesting that alternative treatments of rainfall – different lag structures or cumulative anomalies – might emphasise associations our monthly summaries did not.

Since the outbreak in 2022, which included the first detections of JEV in temperate Australia, the virus has been detected again in southeastern Australia in late Spring early Summer–relatively early in the arbovirus season–although it is unknown whether the virus was reintroduced from northern latitudes or had overwintered in the region. The sporadic nature of the more recent years contrasts with the widespread transmission observed in 2022. Previous JEV incursions into far northern Australian mainland were thought to have resulted from separate reintroductions from the Torres Strait [118,119], and the identification of different virus genotypes supports this hypothesis. The closely related flavivirus MVEV has also recurred multiple times in temperate Australia since a significant outbreak in 1974. It is postulated that MVEV exists in enzootic circulation in northern Australia before being reintroduced into southern latitudes with La Niña-associated climatic patterns [120–122]. Our vector and host suitability models are suggestive of at least moderately suitable habitat in the tropical north throughout the temperate winter, though early detections in temperate Australia weigh against a simple narrative of spread from tropical north to temperate south. Instead, an uncharacterised mechanism may have facilitated overwintering of the virus in temperate Australia, as has been suggested in endemic regions of Asia [118]. Clearly, the mechanisms, hosts and vectors that could potentially reintroduce the virus or support its overwintering in the

Australian context are subject to conjecture and require further investigation (e.g., [119,123], and [87] for MVEV). Overall, evidence since 2022 indicates some similarity with epidemiology in the established range of JEV in Southeast Asia, which has been characterised as endemic in tropical latitudes, seasonally epidemic in temperate latitudes, and somewhere in between in the sub-tropical areas [118].

The difference in scale of realised transmission from 2022–2023 illustrates the limits of correlative habitat models. Environmental conditions were similarly favourable to the preceding year and surveillance effort was enhanced yet no JEV outbreak materialised (though there was an major outbreak of MVE in Australia [122]). Our models describe where conditions are suitable, not whether transmission will occur, or amplify to detectable or epidemic levels. Mechanistic models incorporating vector, host, and immunological dynamics are needed to explore why some years produce outbreaks and others do not [124], and emerging long-term research tracking waterbird movements through inland drainage basins [112,125] may help illuminate how hosts connect favourable habitats across the landscape. Identifying whether JEV is now persisting in Australia, what distinguishes outbreak years, and elucidating the dynamics of co-circulating flaviviruses remains a critical research priority.

Our predictions of summer habitat suitability for *Culex* vector species differ in spatial emphasis from those of Furlong et al [18], the only other published habitat models for Australian JEV vectors. While both models predicted broad suitability over many land cover types, the Furlong et al. individual species models for *Cx annulirostris*, *Cx quinquefasciatus* and *Cx sitiens* indicated strongest suitability on the coastal margins whereas our cohort model emphasised suitable habitat across northern Australia and through the major inland drainage basins. The difference between our cohort model and a *Cx sitiens* only model is expected as that species is mostly associated with saline coastal fringe habitats [23], but the difference between our model and the *Cx annulirostris* and *Cx quinquefasciatus* models of Furlong et al. is striking and has potential public health significance: Australia's population is concentrated on the coastal margin, particularly in the south east, so a coastal emphasis in JEV vector habitat would imply a larger population at risk than our inland focused predictions suggest.

The divergence between our predictions and those of Furlong et al. likely reflects differences in training data and bias mitigation rather than the differences between single species and cohort modelling approaches. Our dataset incorporated mosquito surveillance records that expanded the training data approximately 20-fold beyond the publicly available biodiversity atlas records used by Furlong et al., including substantially more records from inland and remote locations (see S8 Fig) that likely better represent the environmental covariate space these species occupy. The volume and temporal depth of the additional records also allowed us to fit models to time-varying predictors rather than long-term synoptic summaries, capturing transient high-suitability conditions that static approaches cannot represent. We also employed different approaches to neutralising spatial bias in the occurrence datasets compared to Furlong et al. We supplied our models with target-group background points from Kingdom Animalia that shared a similar accessibility bias signature to our occurrence data, following [90,126,127]. Furlong et al. supplied an accessibility layer to weight background samples. In the language of maxent's creators the bias layer modifies the uniform spatial prior [128], though its effect can also be thought of as acting as an offset term representing exposure [129]. Our approach and that of Furlong and colleagues are similar in intent. Neither approach can create signal of habitat suitability in areas with environmental conditions poorly represented in, or absent from, the training sample and it remains to be seen whether our approach to mitigating spatial bias would have produced outputs with less apparent bias given the training data that Furlong and colleagues accessed.

Spatial bias toward population centres in biodiversity atlas records is typically severe, and particularly so for entomological collections [95,96,130]. Meanwhile standard model evaluation metrics are generally unable to indicate distortions due to sample bias [95]. Sampling bias can itself be plausibly correlated with habitat quality since human population centres or access infrastructure and suitable habitat for organisms have similar associations in environmental space [93]. Where spatial bias in samples is consistently correlated with environmental drivers resulting niche models can manifest those potentially spurious patterns across landscape scales [131]. More structured, ecologically stratified vector surveys remain

the most robust long-term solution [132]. In the interim, researchers can examine, transparently report, and attempt to neutralise bias using established methods [127,133], as we have endeavoured to do here.

These considerations extend to downstream analyses that build on vector distribution models. Skinner et al. [134] recently published an assessment of JEV spillover risk to the Australian population that incorporated mechanistic elements of vector feeding preferences and host contact rates—a valuable conceptual advance. However, that analysis took the vector distributions from Furlong et al. as an input, so any spatial bias in the underlying habitat suitability estimates would propagate into the spillover risk surface. Our results suggest caution in interpreting modelled transmission risk derived from strongly biased public occurrence databases, and highlight the value of extracting additional records from other sources where available and flagging potential bias in risk estimates intended to inform public health decision-making.

## Combining host and vector habitat suitability models

Our use of an empirical cumulative distribution function to combine the response scale of the target-group background model for vectors with the presence absence model for ardeid hosts is a notable contribution to the habitat suitability literature. Integrated distribution models are increasingly applied to combine heterogeneous datasets for a single target species [93,135,136], but our approach extends this logic to combine functionally distinct ecological groups into a representation of the fundamental reservoir required for pathogen transmission. The resulting surface, based on the minimum rescaled suitability of vectors and hosts, provides a principled way to identify areas where both components of the transmission cycle coincide. Walsh and colleagues [19] produced an alternative geospatial model of JEV suitability in Australia using the same detection dataset but different modelling approaches and covariates. While both models indicate relatively high suitability inland of the Great Dividing Range—coinciding with the epicentre of the 2022 outbreak—the model of Walsh et al. also suggested very high suitability along southern and eastern coastal margins. The relative absence of detected human infections in those densely populated areas, despite presumably greater opportunity for detection, weighs against that emphasis.

## Limitations and further work

### Data limitations

The JEV locations used to rescale our combined surface were obtained from public domain sources and represent only a subset of confirmed infections and locations. We lacked information on spatial precision or expert confidence in attributed locations and extracted single point values from corresponding raster cells in creating the empirical cumulative distribution functions. Bootstrapping or areal summaries might have been a more appropriately conservative approach to represent location uncertainty. Critically, because we used these points to rescale the component layers, they cannot be used for independent validation of the combined surface. A more comprehensive dataset incorporating detections across multiple host species, with appropriate characterisation of surveillance effort, would enable more robust estimation and evaluation. Such work is underway in collaboration with relevant agencies.

The mosquito occurrence dataset we used was compiled under time pressure during an active outbreak. We were only able to model occurrence rather than abundance, though more dynamic models would be attainable if reliable abundance data were available at national scale (e.g., [137]). Critically, abundance data support estimation of important parameters influencing transmission such as the force of infection from mosquitoes to hosts, or vectorial capacity [138]. Our post-hoc comparison suggested good correspondence between predicted suitability and observed abundance per calendar month in temperate and sub-tropical latitudes, but poorer correspondence in the tropical north where predictions missed nuance in seasonal abundance patterns. In the subset of data with abundance attributes, interpretation was complicated by varying trapping methods, counting protocols, and jurisdictional differences in data management. A coherent approach and protocols for collating and harmonising mosquito abundance data would be an important next step for future outbreak preparedness.

## Model limitations

Our target-group background of Animalia records reflected a similar pattern of accessibility bias to the mosquito surveillance network, but accessibility (travel time to a large city) is only part of the bias picture. Surveillance networks do distribute traps according to human population centres but strategies may also reflect allocation according to past detections, transmission risk factors, vulnerable populations, and likely larval sources. These nuances are not captured by general accessibility metrics. Future modelling could invest more effort in background sampling that more closely reflects the specific bias signature of mosquito surveillance.

Our temporally dynamic models, while capturing month-to-month variation in environmental suitability, include no mechanistic representation of population dynamics, dispersal, breeding phenology, or immunity. Temporal variability was implicitly modelled as covariate combinations with no consideration of residence time or movement scales in geographic space. As mentioned earlier such correlative approaches cannot explain why the 2022–2023 summer, with similarly favourable environmental conditions, produced only relatively isolated detections. Mechanistic transmission models incorporating vector, host, and immunological dynamics would allow exploration of hypotheses regarding the stochastic nature of JEV risk in Australia; several have been published for JEV (reviewed in [124]) though to date they have not included realistic spatial variation. Integrating mechanistic and distributional approaches should be a priority for future work.

## Host species scope

We excluded feral pigs from our host models due to the absence of reliable distribution data at the time of analysis. Given the numerous detections in feral pigs during 2022, their capacity for high viraemia [55,118], and reported abundance and extent [36], there were conceptual grounds for their inclusion. Where feral pigs and ardeid birds share habitat preferences their exclusion would not be consequential, but it may have contributed to us underestimating host suitability in wooded habitats in Australia's tropical north where feral pigs are reportedly abundant but ardeid density may be lower (though modelled suitability for the *Culex* was also low in those habitats, which would limit changes to the combined model). On the other hand, since pigs are not thought likely to contribute to the transmission of MVEV (reviewed in [139]), our models with only waterbird hosts may offer greater flavivirus generality by virtue of that absence. A recently published national feral pig distribution model [140] could be incorporated into future JEV assessments.

## Conclusion

Our time-varying geospatial models of habitat suitability for vectors and vertebrate hosts for JEV in Australia were based on the most comprehensive national datasets for ardeid hosts and mosquito vectors available. They illuminate the temporal variation in background conditions that may lead to spillover transmission of JEV to humans. Such models have demonstrated utility in public health contexts, and the models, datasets and insights generated through this project could be further leveraged to review and improve preparedness for surveillance to avoid future outbreaks. Our model representation of the fundamental habitat for vectors and vertebrate hosts for JEV in Australia could be combined with location data on all JEV infections in mosquitoes, humans, and domestic and feral pigs, and other surveillance resources, to provide a more nuanced and direct examination of JEV transmission risk.

## Supporting information

**S1 Fig. Map of Australian mainland and nearby islands featuring modified Koppen climate classification, major drainage basins, the Great Dividing Range, areas with population density exceeding 1000 people per square kilometre, and the approximate location of JEV detections in the public domain.** Base maps obtained from Australian Bureau of Statistics (CC BY 4.0) https://www.abs.gov.au/statistics/standards/australian-statistical-geography-standard-asgs-edition-3/jul2021-jun2026/access-and-downloads/digital-boundary-files. (TIF)

**S2 Fig. Distribution of occurrence data for the *Culex* species included in our habitat suitability model for the cohort.** Black points indicate records obtained from Australian state and territory surveillance and published literature and databases. Red points were obtained from Global Biodiversity Information Facility and Atlas of Living Australia. Points are semi-transparent (alpha = 0.25). Those that appear solid result from accumulation of at least four records at one location. Base maps obtained from Australian Bureau of Statistics (CC BY 4.0) https://www.abs.gov.au/statistics/standards/australian-statistical-geography-standard-asgs-edition-3/jul2021-jun2026/access-and-downloads/digital-boundary-files.
(TIF)

**S3 Fig. Distribution of occurrence data for the ardeid species included in our habitat suitability model for the cohort.** Points are semi-transparent (alpha = 0.1). Those that appear solid result from accumulation of at least 10 records at one location. Base maps obtained from Australian Bureau of Statistics (CC BY 4.0) https://www.abs.gov.au/statistics/standards/australian-statistical-geography-standard-asgs-edition-3/jul2021-jun2026/access-and-downloads/digital-boundary-files.
(TIF)

**S4 Fig. Global correlation between environmental and land cover variables used in the habitat suitability models for ardeid hosts and *Culex* vectors.**
(TIF)

**S5 Fig. Distribution of values for selected proportional land cover and environmental variables over the Australian mainland and nearby islands.** The values correspond to a 1 km grid. Values for February 2022 are displayed for the temporally dynamic environmental variables (Night T C, Greenness (EVI), Wetness (TCW) and Rainfall (mm)). Base maps obtained from Australian Bureau of Statistics (CC BY 4.0) https://www.abs.gov.au/statistics/standards/australian-statistical-geography-standard-asgs-edition-3/jul2021-jun2026/access-and-downloads/digital-boundary-files.
(TIF)

**S6 Fig. Cumulative proportion of Animalia and selected *Culex* records by month of observation between 2000 and June 2022.**
(TIF)

**S7 Fig. Spatial blocks used for model cross validation.** Each block is labelled with the cross-validation fold to which it was randomly assigned subject to the overall criterion of approximate balance of presence and target background points per fold. Base maps obtained from Australian Bureau of Statistics (CC BY 4.0) https://www.abs.gov.au/statistics/standards/australian-statistical-geography-standard-asgs-edition-3/jul2021-jun2026/access-and-downloads/digital-boundary-files.
(TIF)

**S8 Fig. Accessibility (least cost travel time to city of more than 50K population) for *Culex* records in GBIF and Atlas of Living Australia, *Culex* records from surveillance trap locations, Animalia 'target group background' occurrences from GBIF, and a random sample of accessibility values from the desired prediction area of mainland Australia and nearby islands.**
(TIF)

**S1 File. Animation.** Monthly predictions of habitat suitability for *Culex* cohort to July 2021–June 2023 available at this link. B**ase maps obtained from Australian Bureau of Statistics (CC BY 4.0)** https://www.abs.gov.au/statistics/standards/australian-statistical-geography-standard-asgs-edition-3/jul2021-jun2026/access-and-downloads/digital-boundary-files.
(GIF)

**S9 Fig. Overview of input occurrence data and penalised logistic regression model of relative habitat suitability for *Culex annulirostris* in Australia.** A) the distribution of the combined occurrence data for *Culex annulirostris*; B) the distribution of background occurrence data for Kingdom Animalia, including any trap sites where other mosquitoes but not *Cx annulirostris* were trapped; C) the relative 'importance' of top seven covariates as calculated by permutation on ensemble fit; D) prediction of relative habitat suitability to the outbreak period (February 2022); E–G) the shape of the marginal relationship between relative habitat suitability and the three most influential predictor variables; H–J) prediction of relative habitat suitability from December 2021 and June 2022, and standard deviation of predictions across the temporal sequence from July 2021 to June 2023. Base maps for A, B, and H–J obtained from Australian Bureau of Statistics (CC BY 4.0) https://www.abs.gov.au/statistics/standards/australian-statistical-geography-standard-asgs-edition-3/jul2021-jun2026/access-and-downloads/digital-boundary-files.
(TIF)

**S10 Fig. Evaluation plots of vector habitat suitability model according to the month of occurrence record and latitude band.** A) habitat suitability predictions at raster cells corresponding to *Cx* cohort occurrence data, and, B) independent comparison of maximum abundance per cell *x* month from the subset of those cells where the occurrence data from trap collections had an abundance attribute. Outliers (extreme high values) were omitted from B to emphasise the core 95% of the distribution.
(TIF)

**S2 File. Animation.** Monthly predictions of habitat suitability for ardeid cohort to July 2021–June 2023 available at this link. **Base maps obtained from Australian Bureau of Statistics (CC BY 4.0)** https://www.abs.gov.au/statistics/standards/australian-statistical-geography-standard-asgs-edition-3/jul2021-jun2026/access-and-downloads/digital-boundary-files.
(GIF)

**S11 Fig. Sample empirical cross-validation of the ardeid host habitat suitability model to October 2022 with maximum abundance in that month obtained from independent dataset of aerial survey records from the Australian Waterbird Survey [105].** Grey dots indicate sightings of non-ardeid species and also indicate the flight path; black circles correspond to ardeid species sightings and are scaled by the recorded number of individuals. Base maps obtained from Australian Bureau of Statistics (CC BY 4.0) https://www.abs.gov.au/statistics/standards/australian-statistical-geography-standard-asgs-edition-3/jul2021-jun2026/access-and-downloads/digital-boundary-files.
(TIF)

**S3 File. Animation. Monthly predictions of habitat suitability for *Cx. annulirostris* to July 2021–June 2023** available at this link. Base maps obtained from Australian Bureau of Statistics (CC BY 4.0) https://www.abs.gov.au/statistics/standards/australian-statistical-geography-standard-asgs-edition-3/jul2021-jun2026/access-and-downloads/digital-boundary-files.
(GIF)

## Acknowledgments

We thank the many local governments and personnel who support mosquito-based surveillance activities in Australia. The Communicable Disease Incident of National Significance, declared by the funder in 2022, facilitated the request for access to mosquito surveillance data used in this study. The dataset we collated for this project featuring those collections was made possible by members of the Medical Entomology Team of Western Australia's Environmental Health Directorate, Public Health Entomology, Department of Health Victoria, Medical Entomology Team of Northern Territory Government, NSW Mosquito Monitoring and Arbovirus Surveillance Program, and the Health Protection and Regulation Branch and

Public Health Units, Queensland Health. Shannon Melody from the Tasmanian Department of Health shared data from a mosquito surveillance trial in northern Tasmania, Boni Sebayang provided primary mosquito trap data, and Alex Potter of the Smithsonian Institute helped with initial extraction of relevant mosquito records from the VectorMap database. On the ardeid host side, we are grateful for the diligent efforts of countless bird enthusiasts, as rendered accessible through eBird and Birdlife Australia's birdata platform. Joris Driessen (BirdLife Australia) extracted ardeid records and other complete checklists from birdata on our behalf. Jen Rozier (Malaria Atlas Project) facilitated access to MODIS data products post-processed to fill gaps introduced by cloud cover.

Thanks to the members and invited experts of the National Arbovirus and Malaria Advisory Committee, and Wildlife Health Australia for sharing their expertise and ideas in workshops convened during the outbreak. James McCaw contributed to the modelling strategy. Earlier drafts of this manuscript were improved by comments from David Price, Jennifer Flegg, and Gerry Ryan.

## Author contributions

**Conceptualization:** Nick Golding, Freya M Shearer.

**Data curation:** David H Duncan, Lucinda E Harrison, Abbey Potter, Craig Brockway, Kimberly L Miller, Stephen L Doggett, Rebecca Feldman, Peter J Neville, Andrew F van den Hurk, Cassie C Jansen, Michaela Hobby, Vicki Burns, Nina Kurucz.

**Formal analysis:** David H Duncan, Nick Golding.

**Funding acquisition:** Nick Golding, Freya M Shearer.

**Investigation:** David H Duncan, Lucinda E Harrison, Abbey Potter, Craig Brockway, Kimberly L Miller, Stephen L Doggett, Rebecca Feldman, Peter J Neville, Andrew F van den Hurk, Cassie C Jansen, Andrew Vickers, Nina Kurucz, Nick Golding, Freya M Shearer.

**Methodology:** David H Duncan, Lucinda E Harrison, Nick Golding, Freya M Shearer.

**Project administration:** David H Duncan, Freya M Shearer.

**Resources:** Stephen L Doggett, Rebecca Feldman, Peter J Neville, Cassie C Jansen, Andrew Vickers, Nina Kurucz, Nick Golding, Freya M Shearer.

**Software:** David H Duncan, Lucinda E Harrison, Nick Golding, Freya M Shearer.

**Supervision:** Nick Golding, Freya M Shearer.

**Validation:** David H Duncan, Lucinda E Harrison.

**Visualization:** David H Duncan, Lucinda E Harrison, Nick Golding, Freya M Shearer.

**Writing – original draft:** David H Duncan.

**Writing – review & editing:** David H Duncan, Lucinda E Harrison, Abbey Potter, Craig Brockway, Kimberly L Miller, Stephen L Doggett, Rebecca Feldman, Peter J Neville, Andrew F van den Hurk, Cassie C Jansen, Michaela Hobby, Vicki Burns, Andrew Vickers, Nina Kurucz, Nick Golding, Freya M Shearer.

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
