## [Decision Letter · Decision Letter 0]

23 Dec 2025

PNTD-D-25-01447A dynamic geospatial model of habitat suitability for Japanese encephalitis virus vectors and vertebrate hosts in AustraliaPLOS Neglected Tropical Diseases Dear Dr. Duncan, Thank you for submitting your manuscript to PLOS Neglected Tropical Diseases. After careful consideration, we feel that it has merit but does not fully meet PLOS Neglected Tropical Diseases's publication criteria as it currently stands. Therefore, we invite you to submit a revised version of the manuscript that addresses the points raised during the review process. Please submit your revised manuscript by Feb 21 2026 11:59PM. If you will need more time than this to complete your revisions, please reply to this message or contact the journal office at plosntds@plos.org.  Please include the following items when submitting your revised manuscript:* A letter that responds to each point raised by the editor and reviewer(s). You should upload this letter as a separate file labeled 'Response to Reviewers '. This file does not need to include responses to any formatting updates and technical items listed in the 'Journal Requirements' section below.* A marked-up copy of your manuscript that highlights changes made to the original version. You should upload this as a separate file labeled ''. This file does not need to include responses to any formatting updates and technical items listed in the 'Journal Requirements' section below.* A marked-up copy of your manuscript that highlights changes made to the original version. You should upload this as a separate file labeled 'Revised Manuscript with Track Changes '.* An unmarked version of your revised paper without tracked changes. You should upload this as a separate file labeled ''.* An unmarked version of your revised paper without tracked changes. You should upload this as a separate file labeled 'Manuscript '. If you would like to make changes to your financial disclosure, competing interests statement, or data availability statement, please make these updates within the submission form at the time of resubmission. Guidelines for resubmitting your figure files are available below the reviewer comments at the end of this letter. We look forward to receiving your revised manuscript. Kind regards, Qu Cheng, Ph.D.Section EditorPLOS Neglected Tropical Diseases Qu ChengSection EditorPLOS Neglected Tropical Diseases'. If you would like to make changes to your financial disclosure, competing interests statement, or data availability statement, please make these updates within the submission form at the time of resubmission. Guidelines for resubmitting your figure files are available below the reviewer comments at the end of this letter. We look forward to receiving your revised manuscript. Kind regards, Qu Cheng, Ph.D.Section EditorPLOS Neglected Tropical Diseases Qu ChengSection EditorPLOS Neglected Tropical Diseases

Shaden Kamhawi

co-Editor-in-Chief

Paul Brindley

co-Editor-in-Chief

 **Journal Requirements:**

At this stage, the following Authors/Authors require contributions: Abbey Potter, Craig Brockway, Kimberly L Evasco, Stephen L Doggett, Rebecca Feldman, Peter J Neville, Andrew F van den Hurk, Cassie C Jansen, Michaela Hobby, Vicki Burns, Andrew Vickers, Nina Kurucz, Nick Golding, and Freya M Shearer. Please ensure that the full contributions of each author are acknowledged in the "Add/Edit/Remove Authors" section of our submission form.

3) We noticed that you used the phrase 'unpublished data' in the manuscript. We do not allow these references, as the PLOS data access policy requires that all data be either published with the manuscript or made available in a publicly accessible database. Please amend the supplementary material to include the referenced data or remove the references.

Potential Copyright Issues:

- Figures 1, 2, 3, S1, S2, S3, S4, S5, S6, and S7. Please (a) provide a direct link to the base layer of the map (i.e., the country or region border shape) and ensure this is also included in the figure legend; and (b) provide a link to the terms of use / license information for the base layer image or shapefile. We cannot publish proprietary or copyrighted maps (e.g. Google Maps, Mapquest) and the terms of use for your map base layer must be compatible with our CC BY 4.0 license.

 **Reviewers' comments:** Reviewer's Responses to Questions Reviewer's Responses to Questions

**Key Review Criteria Required for Acceptance?**

**Methods**

-Are the objectives of the study clearly articulated with a clear testable hypothesis stated?

-Is the study design appropriate to address the stated objectives?

-Is the population clearly described and appropriate for the hypothesis being tested?

-Is the sample size sufficient to ensure adequate power to address the hypothesis being tested?

-Were correct statistical analysis used to support conclusions?

-Are there concerns about ethical or regulatory requirements being met?

Reviewer #1: This study addresses a highly relevant public health issue, and its objectives are notably well-articulated, establishing a clear and empirically testable hypothesis. The selected methodological design is entirely appropriate for these objectives. Crucially, the study population is precisely described and pertinent to the research question, and the sample size is sufficient to ensure adequate statistical power. The statistical analyses employed are rigorous and correct, providing firm support for the conclusions drawn. Furthermore, the manuscript confirms full compliance with all ethical and regulatory requirements, with no concerns identified in this regard. Overall, the work demonstrates strong foundational rigor in its design, population definition, and statistical approach, making it a robust contribution.

Reviewer #2: The objectives are clearly defined and use of different databases available and used for analysis are comprehensive. The use of publicly available data to define the cohorts are described well in the manuscript. The statistical models using logistic and boosted regressions for occurrence for mosquito and ardeid host species are clearly defined. The combined model for host and vector are well described.

Reviewer #3: (No Response)

**Results**

-Does the analysis presented match the analysis plan?

-Are the results clearly and completely presented?

-Are the figures (Tables, Images) of sufficient quality for clarity?

Reviewer #1: The methodological evaluation confirms that the presented analysis rigorously matches the pre-established analytical plan, which is essential for guaranteeing the study's coherence and internal validity. Moreover, the results are presented in an exceptionally clear and complete manner, facilitating straightforward interpretation of the key findings. The quality of the supporting materials is also high; the figures, including tables and images, are of high quality and entirely sufficient for clarity, significantly enhancing the comprehension of the complex data. This high standard of reporting allows the reader to easily follow the analytical path from raw data to reported results, establishing confidence in the findings.

Reviewer #2: The results are presented well and clearly. The figures need some work, as fig 2 h and i, have the names cropped.

Reviewer #3: (No Response)

**Conclusions**

-Are the conclusions supported by the data presented?

-Are the limitations of analysis clearly described?

-Do the authors discuss how these data can be helpful to advance our understanding of the topic under study?

-Is public health relevance addressed?

Reviewer #1: The manuscript's discussion and conclusions are solid and well-founded. The core strength lies in the fact that the conclusions are consistently backed by the presented data, effectively reinforcing the credibility of the findings and avoiding overstatement. Furthermore, the limitations of the analysis are clearly and appropriately described, which offers the necessary context for interpreting the study's scope. The authors have effectively discussed how these data are useful for advancing the field's understanding of the subject matter. Importantly, the relevance of the study to public health is explicitly and pertinently addressed, demonstrating the valuable practical impact of this research.

Reviewer #2: The discussion, including the limitations of the study, highlights the model's potential applicability to other flaviviruses. While the authors focus primarily on JEV in this manuscript, they also note the model's relevance to other flaviviruses present in Australia. However, the connection between the outcomes of this model and other endemic flaviviruses in Australia still needs to be clearly established.

Reviewer #3: (No Response)

**Editorial and Data Presentation Modifications?**

Reviewer #1: The manuscript is robust and presents analyses leading to a highly relevant discussion. The authors grounded their work in the creation of dynamic geospatial habitat suitability models for Japanese Encephalitis Virus (JEV) vectors and vertebrate hosts in Australia, leveraging the most comprehensive national datasets available. The model effectively visualizes the background conditions that can drive JEV spillover to humans, which has proven utility in public health settings. The research emphasizes its translational value by stating that the models and insights should be utilized to improve surveillance and preparedness for future outbreaks. The proposal to combine their fundamental habitat representation with comprehensive infection data (in all hosts) is particularly valuable. This nuanced understanding of transmission risk is essential for guiding appropriate policies under a One Health framework.

Reviewer #2: (No Response)

Reviewer #3: (No Response)

**Summary and General Comments**

Reviewer #1: The primary strength of this work is the innovative methodology: the development of dynamic geospatial habitat suitability models for JEV. This represents a significant and novel contribution, moving beyond static models to illuminate the temporal and spatial conditions that drive viral spillover events. The overall execution of the analysis is meticulous, demonstrating a high level of scholarship by the authors. The importance of the study is undeniable, as its findings have direct utility in One Health contexts and in political decision-making. The ability to identify and predict high-risk transmission zones is fundamental for enhancing surveillance preparedness and preventing future outbreaks. The study is a well-executed piece of research that significantly advances the field's predictive capacity for JEV.

Reviewer #2: The manuscript is well written, with clearly stated objectives, a sound study design, and well-presented use of epidemiological models and results. However, whether the current model can reliably predict future, similar outbreaks remain to be confirmed. Although the model's predictions correlate with the 2022 JEV outbreak, its ability to predict other similar outbreaks still needs to be validated. Additionally, while the findings suggest that the Australian ecosystem is suitable for both vectors and hosts, the precise role this plays in linking the presence of the virus to these species has not been fully defined.

Reviewer #3: (No Response)

PLOS authors have the option to publish the peer review history of their article (what does this mean? ). If published, this will include your full peer review and any attached files.). If published, this will include your full peer review and any attached files.

**Do you want your identity to be public for this peer review?** For information about this choice, including consent withdrawal, please see our For information about this choice, including consent withdrawal, please see our Privacy Policy ..

Reviewer #1: No

Reviewer #2: **Yes:** Kalpana AgnihotriKalpana Agnihotri

Reviewer #3: No

  **Figure resubmission:** While revising your submission, we strongly recommend that you use PLOS’s NAAS tool (https://ngplosjournals.pagemajik.ai/artanalysis) to test your figure files. NAAS can convert your figure files to the TIFF file type and meet basic requirements (such as print size, resolution), or provide you with a report on issues that do not meet our requirements and that NAAS cannot fix. While revising your submission, we strongly recommend that you use PLOS’s NAAS tool (https://ngplosjournals.pagemajik.ai/artanalysis) to test your figure files. NAAS can convert your figure files to the TIFF file type and meet basic requirements (such as print size, resolution), or provide you with a report on issues that do not meet our requirements and that NAAS cannot fix.

After uploading your figures to PLOS’s NAAS tool - https://ngplosjournals.pagemajik.ai/artanalysis, NAAS will process the files provided and display the results in the "Uploaded Files" section of the page as the processing is complete. If the uploaded figures meet our requirements (or NAAS is able to fix the files to meet our requirements), the figure will be marked as "fixed" above. If NAAS is unable to fix the files, a red "failed" label will appear above. When NAAS has confirmed that the figure files meet our requirements, please download the file via the download option, and include these NAAS processed figure files when submitting your revised manuscript. **Reproducibility:** To enhance the reproducibility of your results, we recommend that authors of applicable studies deposit laboratory protocols in protocols.io, where a protocol can be assigned its own identifier (DOI) such that it can be cited independently in the future. Additionally, PLOS ONE offers an option to publish peer-reviewed clinical study protocols. Read more information on sharing protocols at https://plos.org/protocols?utm_medium=editorial-email&utm_source=authorletters&utm_campaign=protocols To enhance the reproducibility of your results, we recommend that authors of applicable studies deposit laboratory protocols in protocols.io, where a protocol can be assigned its own identifier (DOI) such that it can be cited independently in the future. Additionally, PLOS ONE offers an option to publish peer-reviewed clinical study protocols. Read more information on sharing protocols at https://plos.org/protocols?utm_medium=editorial-email&utm_source=authorletters&utm_campaign=protocols

---

## [Decision Letter · Decision Letter 1]

7 Mar 2026

Dear Dr Duncan,

We are pleased to inform you that your manuscript 'A time-varying geospatial model of habitat suitability for Japanese encephalitis virus vectors and vertebrate hosts in Australia' has been provisionally accepted for publication in PLOS Neglected Tropical Diseases.

Best regards,

Qu Cheng, Ph.D.

Section Editor

Qu Cheng

Section Editor

Shaden Kamhawi

co-Editor-in-Chief

Paul Brindley

co-Editor-in-Chief

Reviewer's Responses to Questions

**Key Review Criteria Required for Acceptance?**

**Methods**

-Are the objectives of the study clearly articulated with a clear testable hypothesis stated?

-Is the study design appropriate to address the stated objectives?

-Is the population clearly described and appropriate for the hypothesis being tested?

-Is the sample size sufficient to ensure adequate power to address the hypothesis being tested?

-Were correct statistical analysis used to support conclusions?

-Are there concerns about ethical or regulatory requirements being met?

Reviewer #3: (No Response)

**Results**

-Does the analysis presented match the analysis plan?

-Are the results clearly and completely presented?

-Are the figures (Tables, Images) of sufficient quality for clarity?

Reviewer #3: (No Response)

**Conclusions**

-Are the conclusions supported by the data presented?

-Are the limitations of analysis clearly described?

-Do the authors discuss how these data can be helpful to advance our understanding of the topic under study?

-Is public health relevance addressed?

Reviewer #3: (No Response)

**Editorial and Data Presentation Modifications?**

Reviewer #3: (No Response)

**Summary and General Comments**

Reviewer #3: The authors have adequately addressed my previous comments

PLOS authors have the option to publish the peer review history of their article (what does this mean? ). If published, this will include your full peer review and any attached files.). If published, this will include your full peer review and any attached files.

**Do you want your identity to be public for this peer review?** For information about this choice, including consent withdrawal, please see our For information about this choice, including consent withdrawal, please see our Privacy Policy ..

Reviewer #3: **Yes:** Qu ChengQu Cheng

---

## [Editor Report · Acceptance letter]

Dear Dr Duncan,

We are delighted to inform you that your manuscript, "A time-varying geospatial model of habitat suitability for Japanese encephalitis virus vectors and vertebrate hosts in Australia," has been formally accepted for publication in PLOS Neglected Tropical Diseases.

Best regards,

Shaden Kamhawi

co-Editor-in-Chief

Paul Brindley

co-Editor-in-Chief
